# Management of possible serious bacterial infection in young infants closer to home when referral is not feasible: Lessons from implementation research in Himachal Pradesh, India

**Nidhi Goyal[1], Temsunaro Rongsen-Chandola[1]\*, Mangla Sood[2], Bireshwar Sinha[1], Amit Kumar[1], Shamim Ahmad Qazi[3], Samira Aboubaker[3], Yasir Bin Nisar[3], Rajiv Bahl[3], Maharaj Kishan Bhan[4,5†], Nita Bhandari[1]**

1 Centre for Health Research and Development, Society for Applied Studies, Kalu Sarai, New Delhi, India,
2 Department of Pediatrics, Indira Gandhi Medical College, Shimla and Child Health, National Health Mission, Himachal Pradesh, India, 3 Department of Maternal, Newborn, Child and Adolescent Health and Ageing, World Health Organization, Geneva, Switzerland, 4 Indian Institute of Technology-Delhi, New Delhi, India, 5 Knowledge Integration and Translational Platform (KnIT), Biotechnology Industry Research Assistance Council (BIRAC), New Delhi, India

† Deceased.
* naro@sas.org.in

## Abstract

### Background

Government of India and the World Health Organization have guidelines for outpatient management of young infants 0–59 days with signs of Possible Serious Bacterial Infection (PSBI), when referral is not feasible. Implementation research was conducted to identify facilitators and barriers to operationalizing these guidelines.

### Methods

Himachal Pradesh government implemented the guidelines in program settings supported by Centre for Health Research and Development, Society for Applied Studies. The strategy included community sensitization, skill enhancement of Accredited Social Health Activists (ASHA), Auxiliary Nurse Midwives (ANMs) and Medical Officers (MOs) to identify PSBI and treat when referral was not feasible. The research team collected information on facilitators and barriers. A technical support unit provided training and oversight.

### Findings

Among 1997 live births from June 2017 to January 2019, we identified 160 cases of PSBI in young infants resulting in a coverage of 80%, assuming an incidence of 10%. Of these,29 (18.1%) had signs of critical illness (CI), 92 (57.5%) had clinical severe infection (CSI), 5 (3.1%)had severe pneumonia (only fast breathing in young infants 0–6 days), while 34 (21%) had pneumonia (only fast breathing in young infants 7–59 days). Hospital referral

**Data Availability Statement:** The data underlying this study is publicly available at https://figshare.com/articles/dataset/Data_for_PSBI_IR_Manuscript_zip/13325174.

**Funding:** The study was supported by a grant from the Bill and Melinda Gates Foundation (BMGF Grant No. OPP1114815) through the Department of Maternal, Newborn, Child and Adolescent Health and Ageing, World Health Organization, Geneva, Switzerland. TRC received funds through a Technical Service Agreement with WHO, (WHO Reference 2016/640408-0). URL of funder website: www.gatesfoundation.org. The funder, BMGF had no role in study design, data collection and analysis, decision to publish or preparation of the manuscript. SAQ, SA, YBN and RB from WHO were involved in study design, decision to publish and preparation of the manuscript.

**Competing interests:** The authors have declared that no competing interests exist.

was accepted by 48/160 (30%), whereas 112/160 (70%) were treated with the simplified treatment regimens at primary level facilities. Of the 29 infants with CI, 18 (62%) accepted referral; 26 (90%) recovered while 3 (10%) who had accepted referral, died. Of the 92 infants who had CSI, 86 (93%) recovered, 65 (71%) received simplified treatment and one infant who had accepted referral, died. All the five infants who had severe pneumonia, recovered; 3 (60%) had received simplified treatment. Of the 34 pneumonia cases, 33 received simplified treatment of which 5 (15%) failed treatment; two out of these 5 died. Overall, 6/160 infants died (case-fatality-rate 3.4%); 2 in the simplified treatment (case-fatality-rate 1.8%) and 4 in the hospital group (case-fatality-rate 8.3%). Delayed identification and care-seeking by families and health system weaknesses like manpower gaps and interrupted supplies were challenges in implementation.

## Conclusions

Implementation of the guidelines in program settings is possible and acceptable. Scaling up would require creating community awareness, early identification and appropriate care-seeking, strengthening ASHA home-visitation program, building skills and confidence of MOs and ANMs, uninterrupted supplies and a dependable referral system.

## Introduction

Neonatal infections are a leading cause of child mortality, globally [1]. In 2015 there were 696,000 neonatal deaths in India which contributed to 58% of all under-five child deaths [2]. Among these, infection or sepsis accounted for 103,000 deaths and was the second leading cause of neonatal mortality [2]. The recommended treatment of possible serious bacterial infection (PSBI) or sepsis in a young infant <59 days old is referral to a hospital and treatment with injectable antibiotics i.e. gentamicin with either ampicillin or penicillin for at least 7 days [3, 4]. However, hospitalization is not always feasible in low-middle income settings due to various social, logistic, economic and access related issues [3, 5–9]. Studies from Asia and Africa in settings where referral was not feasible have demonstrated that young infants with signs of infection other than critical illness can be treated with simplified treatment regimens, closer to home on an outpatient basis by trained health workers [9–12]. Based on evidence, in 2014 the Government of India (GoI) issued guidelines for the management of sepsis in young infants with a daily injection of gentamicin and twice daily oral amoxicillin for 7 days given by an Auxiliary Nurse Midwife (ANM), a trained village-level health worker based at a Sub Centres (SC), under special situations when inpatient care is not feasible or acceptable to families [3]. In 2015, the World Health Organization (WHO) guidelines for managing PSBI in young infants when referral is not feasible were released [13]. WHO recommended simplified antibiotic regimen treatment, based on the severity of signs and symptoms, closer to home by trained facility-based health workers on an outpatient basis. It also included treating young infants 7–59 days of age with only fast breathing, with oral amoxicillin without referral to a hospital [13].

The GoI guidelines have not been implemented in the country even in places where access to health facilities is challenging. Implementing these would require a) increasing awareness in the community for early identification and care-seeking of sick young infants, b) strengthening the home-visitation program by Accredited Social Health Activist (ASHA), an incentive

based community health worker, in the first 42 days of life to identify children with any sign of PSBI c) capacity building of ANMs. The objective of this project was to encourage the health functionaries in Sangrah, Himachal Pradesh, to use the simplified treatment regimen for young infants with PSBI when referral is not feasible, identify key barriers and facilitators that influence the use of the simplified treatment regimen to achieve 80% identification and treatment of young infants with PSBI with high treatment success and low case fatality and document lessons learnt for subsequent scaling. It was part of a countrywide project conducted in four states in India: Haryana, Himachal Pradesh, Maharashtra and Uttar Pradesh, through the Ministry of Health supported by WHO, to understand how the guidelines can be implemented and scaled up in different contexts.

## Methods

### Study design

Implementation research was conducted with the intent to understand why certain things work or do not work in a given context and use that information to support the process of re-iterative refinement required for successful adaptation and scaling-up of the National guidelines. The implementation strategy was based on the conceptual framework adapted from the Research Effectiveness Adoption Implementation Maintenance RE-AIM framework (Fig 1) [14]. The government health system implemented the guideline and the research team provided support through regular inputs on the fidelity of the implementation and problem identification. Solutions were identified in close collaboration with the government.

**PROJECT STAGE: BEFORE IMPLEMENTATION**

- National and State Level Consultations
  - Government of India guidelines on management of sepsis in young infants under specific situations, 2014; exists but not implemented
  - WHO guidelines on managing possible serious bacterial infection in young infants when referral is not feasible, World Health Organization, issued in 2015
  - Government of India guidelines adapted; addendum issued, 2016
  - No existing implementation model and need to establish demonstration sites

- Technical Support Unit (TSU), comprising of state health officials and researchers, constituted to provide technical oversight, support and encourage ownership of the program by the state
- Synthesis of findings from Situation Analysis and Baseline survey to inform design of implementation strategy

**PROJECT STAGE: DURING IMPLEMENTATION**

- Implementation of the adapted guidelines through the existing health system with minimal external support from research team
- On field support and data collection by research team
  - Monitoring and evaluation of implementation; documentation of adherence to guidelines
  - Program learning to identify gaps and challenges
  - Feedback on facilitators and barriers to program delivery

- Targeted training of health functionaries

- Refining implementation strategy through close collaboration between Government and research team to achieve high coverage of the implementation of adapted guidelines with quality

Problem Identification

Implementation support

Govt led solution design

**PROJECT STAGE: AFTER IMPLEMENTATION**

- Estimate the impact of implementation of the adapted guidelines using the data on PSBI cases
- Dissemination of learning for potential scale up of the intervention
- Scalable implementation strategy

**Fig 1. Conceptual framework for implementing the guidelines on management of PSBI, when referral is not feasible, through the public health system in Sirmaur, Himachal Pradesh** *(adapted from RE-AIM framework).*

## Policy dialogue

At the National level, orientation meetings were held between the Ministry of Health and Family Welfare and experts from WHO to share the findings of relevant research, WHO guidelines for managing PSBI in young infants when referral is not feasible and experience of implementing these guidelines. Thereafter, the Ministry of Health and Family Welfare held a national consultation in October 2016 with technical experts where PSBI management was discussed. This meeting led to a consensus on the need for conducting implementation research, and the participating sites agreed to follow the adapted national guidelines which included using the WHO recommended dose for amoxicillin 100mg/kg/day instead of 50mg/kg/day. It was also agreed that infants aged 7–59 days with only fast breathing be treated with only oral amoxicillin, without referral to the hospital. In June 2017, the government issued an addendum to their 2014 guidelines for the management of sepsis specifying presence of one or more of the following symptoms and signs to identify PSBI i.e. not able to feed, convulsions, fast breathing (60 breaths per minute or more), severe chest indrawing, axillary temperature 37.5˚C or above, axillary temperature less than 35.5˚C and movement only when stimulated or no movement at all [15]. The symptoms of nasal flaring, grunting, skin pustules or boil, blood in stools were disregarded as signs of PSBI in the addendum. The document also emphasized that ANMs and Medical Officers (MO) should confirm the presence of PSBI, initiate and complete treatment for infants with PSBI on out-patient basis, when referral was not feasible.

Several state-level consultative meetings were held in Himachal Pradesh between the research team, state and district government officials and policy makers in the first quarter of 2016 to decide the most relevant area in which to implement the adapted guidelines for management of PSBI, when referral is not feasible. Sangrah block of district Sirmaur was selected, as it is a designated "difficult area" due to its hilly terrain, weather and access and deemed to benefit the most from this implementation research and to generate learnings that could be adapted and applied to other parts of the district and state, for scale up.

## Study area and population

Implementation research was conducted in the medical block of Sangrah, which is one of the five medical blocks in Sirmaur district (https://www.mapsofindia.com/maps/himachal pradesh/himachalpradesh.htm, https://hpsirmaur.nic.in/map-of-district/), having a population of around 77,000 and an annual birth rate of 17/1000 population. Assuming a 10% incidence of PSBI, 218 cases of PSBI were expected over the 20-month study period [10–12]. The block has 128 villages with an average population of 600 in each village (https://hpsirmaur.nic.in/gram-panchayats). Agriculture is the main source of livelihood. Access to health care is difficult due to the terrain, non-availability of ANMs and MOs at the health facilities and high out of pocket expenditure for inpatient care.

## Situation analysis of the study area

A baseline survey was conducted using a mix of qualitative and quantitative methods to understand the knowledge, attitude and practices of the families regarding illnesses in young infants and their care seeking.

Five focus group discussions were conducted with 5–8 participants each that included parents and grandparents of young infants and community elders. During these discussions families of infants who had an illness suggestive of PSBI shared their experiences on the details of the illness, its recognition, time to care seeking, sources from where care was sought and problems, if any, faced in accessing care. Additionally, the team assessed the health facilities using structured forms and conducted in-depth interviews with the care providers, and also

assessed available infrastructure, commodities, services, practices in management of sick young infants and training needs. The learnings were shared with the health authorities and the strategy for the implementation research was formulated with their inputs.

The public health system has two Community Health Centers (CHC) manned by MOs, ANMs, male Multipurpose Health Workers, health supervisors, nurses and pharmacists, nine Primary Health Centers (PHC) helmed by an MO and pharmacist/helper and 27 Sub Centers (SC) that are helmed by ANMs [16]. ASHAs are trained in Home Based Newborn Care (HBNC) and make 6–7 post-natal home visits for newborns in the first 42 days of life [17]. In Sangrah, there are no qualified private practitioners or private hospitals.

The population served by the PHCs and SCs is smaller in Sangrah as compared to the norm in India, because of the difficult terrain [16]. On an average, the nearest PHC is around 30 minutes walking distance from a village and serves a population of 8,000–10,000. Free transportation is provided to mothers and infants through ambulance services under the Janani Swasthya Suraksha Karyakram (JSSK) program [18]. Other health schemes of the Government to promote institutional delivery among poor pregnant women (Janani Suraksha Yojana), training health personnel in basic newborn care and resuscitation (Navjaat Shishu Suraksha Karyakram) and National health protection scheme for hospitalization (Ayushman Bharat Yojana) are also operational [19, 20].

The two CHCs in Sangrah are designated skilled birth attendant delivery points in the block. Around 34% babies are delivered at home [21]. CHC Sangrah, which is about 4–5 hours by road from the farthest part of the block, is the only health centre in the block, providing regular inpatient care services. It has 30 beds but no dedicated pediatric or neonatal beds. 50 bedded sub-district level hospitals in the neighboring blocks of Dadahu and Rajgarh, about an hour away by road on either side of the block, are the closest referral hospitals with pediatric beds. There is a district hospital (300 beds including 20 pediatrics beds, 12 neonatal beds) in the neighboring district of Solan and a medical college hospital (250 bedded, 30 pediatrics beds) in the district headquarters, Nahan, both of which are about 3 hours away by road, on either side of the block. Almost all families of sick young infants seek care at a PHC or CHC, whereas the SCs are generally not utilized as point of care for young infants, most likely because of unpredictable availability of the ANMs.

## Establishment of the Technical Support Unit (TSU)

A TSU comprising of the state health officials, including the Deputy Mission Director—National Health Mission, State Program Officer—Child Health, District Program Officer—Sirmaur, Chief Medical Officer–Sirmaur and the researchers of CHRD-SAS was established and provided inputs on strategy, technical oversight and support for this implementation research. The role of the health authorities was to implement the management of PSBI through the health system and encourage ownership of the program by the state, while the research team's role was to evaluate processes, support skill building and reinforcement, identify gaps in skills, supplies, commodities and challenges. The health officials of the TSU provided relevant trainings to the health functionaries in the block with support from the research team. Members of the TSU communicated at least once in a month, discussed problems identified and provided guidance and possible solutions.

## The research team

The research team, comprising of personnel from CHRD SAS, was trained in project specific activities and worked under the supervision of the TSU. There were four arms of the team that interacted with each other through daily meetings and with the research team members of the TSU through daily and monthly reports and quarterly review meetings.

a. <u>Implementation Support Team</u> provided support to the health system in the implementation of the program by facilitating trainings and community awareness activities and checking the availability of supplies and commodities. They interactedwith the Block Medical Officer (BMO) and the Chief Medical Officer (CMO) frequently, attended and documented the proceedings of the monthly block and district level meetings, provided feedback on the implementation progress, shared the findings of the Program learning team, facilitated training requests and discussed inadequacies in supplies or any other problem newly identified or unresolved.

b. <u>Program Learning Team,</u> trained in qualitative data collection methods, conducted in-depth interviews with families, ASHAs, ANMs and MOs, checked health records maintained by the health functionaries, made direct observations to assess quality, identified training needs and contacted ANMs and MOs to find out new cases of PSBI identified. Their findings were used for further improving the implementation, where ever possible, through discussions with TSU and instituting solutions like targeted trainings.

c. <u>Monitoring and Evaluation Team</u> independently captured information of infants with PSBI from the health facility record and outcome of the illness through interviews with the families. Information was collected in structured forms.

d. <u>Data Management Team</u> managed the data and maintained the database. The on-site programmer was guided and supported by the WHO central data lead.

## Implementation strategy

The research involved identification of symptoms and signs of PSBI in young infants by families and ASHAs and promotion of timely care seeking. The ASHAs used their routine home visitation opportunities for identification of infants with PSBI and directed families to the closest functional health facility. Assessment and confirmation of PSBI and facilitation of referral was by ANMs and MOs at the health centers (SC/PHC/CHC). When referral to hospital was not feasible, simplified antibiotic treatment regimen was initiated.

The details of the implementation strategy, definitions, re-classification of PSBI as critical illness, clinical severe infection, severe pneumonia or pneumonia, and the simplified regimen for management of each classification (Fig 2).

## Data collection instruments and procedures

Both qualitative and quantitative methods of data collection were used. Collection of qualitative data was primarily through in-depth interviews with caregivers of young infants, ASHAs, ANMs, MOs and community leaders, by the program learning team, using semi-structured questionnaires and guides. Information on training, knowledge, practices, and challenges faced was collected from the health functionaries, while information on practices and beliefs around birth and care of newborns and experiences in care seeking was collected from families of sick young infants. Additionally, the team made direct observations.

Information on infants with PSBI was primarily collected through common protocol and forms developed by WHO, that were customized in India and used across all the sites.

The research team used these forms to capture information related to the PSBI cases from the health facility records. The listing of pregnant women, outcomes of the pregnancies and post-natal home visit records were captured from the Reproductive and Child Health (RCH) and HBNC home visitation registers maintained by the ANMs and ASHAs. The research team captured the details of the symptoms and assessment of the sick young infants from the

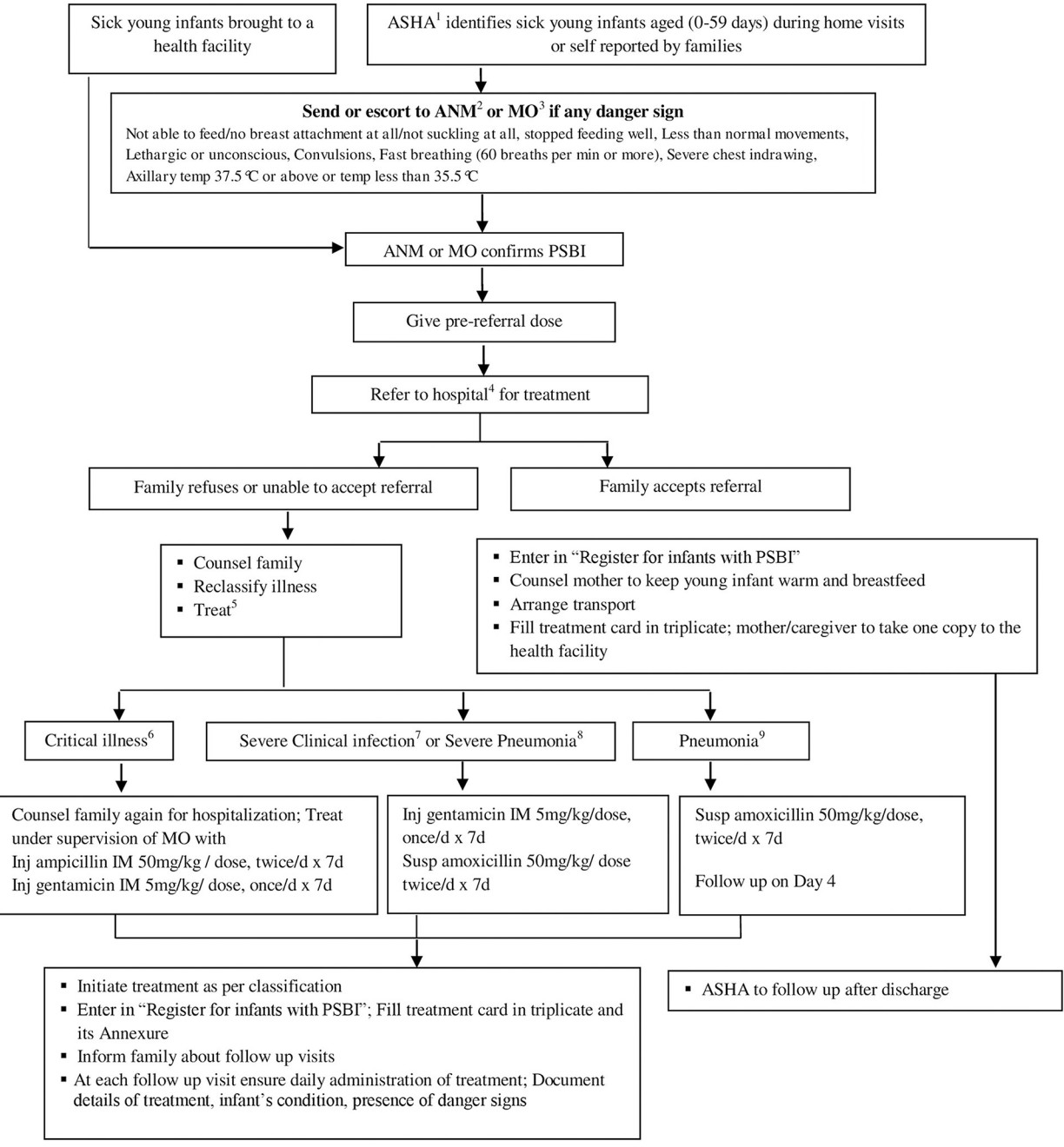

Research Team Activities:
Daily visit to health facility to collect information from records; Day 4-5 home visit for those at home; Take consent, Collect information on care-seeking; Day 8-10 home visit, Conduct interview to collect information on treatment and outcome of illness

[1] Accredited Social Health Activis, an incentive based community health worker
[2] Auxiliary Nurse Midwife, a trained village-level health worker based at a sub-centre
[3] Medical Officer, a qualified medical professional posted at a health centre or hospital
[4] Referred outside the block as there are no hospitals in the block
[5] Treated either as outpatient at the SC/PHC/CHC or as inpatient at a CHC and in a few rare cases at a PHC. The CHC and PHC infrastructure is not equipped for admission of young infants.
[6] Critical illness defined as having any of the following signs: Not able to feed, Convulsions, No movement at all
[7] Clinical severe infection defined as having any of the following signs: Stopped feeding well, movement only when stimulated, severe chest indrawing, Axillary Temp $\geq 37.5°$ C or $< 35.5°$ C
[8] Severe Pneumonia defined as having: Fast Breathing, $\geq 60$/min in young infants aged 0-6 days
[9] Pneumonia defined as having: Fast Breathing, $\geq 60$/min in young infants aged 7-59 days

**Fig 2. Strategy for management of PSBI in sick young infants where referral is not feasible.**

records maintained by the MOs and nurses at the PHCs and CHCs and by the ANMs at the SCs. These details were recorded by the health functionaries at each health centre in a separate register for PSBI infants, the outpatient slip and the IMNCI sick young infants recording forms. The details of referral, treatment regimen and management of PSBI cases were captured from the treatment card and its annexures, filled by the MO, Nurse or ANM at the health centres in triplicate: one copy each for the facility, ASHA and family. The monitoring and evaluation team visited families of the sick young infants between days 8 and 10of the initiation of treatment, on a day convenient to the family, for independent assessment of the course and outcome of illness, after obtaining written informed consent.

## Building health system capacity

**Training.** The training of trainers was conducted by WHO in Delhi in October 2016 where all members of the TSU, including the state health officials and research investigators were trained. A four-day training module, based on the IMNCI module for 0–2 months, was developed specifically, for identification, assessment and management of sick young infants with signs of PSBI including when referral is not feasible. The training included classroom sessions, audio-video based exercises and clinical sessions for hands-on training and practise.

Before the implementation research was initiated, the master trainers; three from the State and four from the research team, conducted 4-day training sessions for all health functionaries (MOs, Ayurvedic Medical Officers, Nurses, ANMs, Supervisors) in the study area, in batches of around 20 each. Separate training sessions were conducted, in batches, for ASHAs on the recognition of danger signs during their home visits, counselling families and timely referral to a closest health facility where personnel trained in management of PSBI were available. Similar training sessions were organized periodically during the period of the study. Additionally, specific targeted training was conducted for MOs, ANMs, and ASHAs throughout the study period, based on identified needs.

**Skill assessment.** The skills of the ASHAs were assessed through direct observations and at home visits where the Program Learning Team accompanied ASHAs. Pre and post training assessments were conducted for ASHAs, ANMs and MOs to assess knowledge, skill retention and further training needs. Identification and referral of PSBI or simple fast breathing can be done earliest by ASHA workers as they are closest to the community and usually the first contact between the SYI and the health system. Evaluation of the accuracy of respiratory rate count in young infants by ASHAs, post training, as compared to a gold standard was conducted and showed acceptable agreement in RR count by Bland-Altman plot, scatter plot and comparison of means of measurements.

**Tools.** Job aids were developed for the ANMs, MOs and for ASHAs to support them in conducting the home visits and deliver targeted messages. These included a flip chart with pictorial representations of danger signs, elements of essential newborn care and messages for care seeking for use by ASHA workers and ANMs; posters with the treatment algorithms and dosage, put up inside the health centres, for easy reference of the MOs and ANMs.

Additionally, the research team provided initial support and guidance on maintaining documents to the health care providers at the health centres. Throughout the implementation the MOs, ANMs and ASHAs were encouraged and motivated to implement the simplified regimen whenever referral was not feasible and share their challenges in managing PSBI cases.

**Community awareness activities.** Messages and tools for community awareness were developed through community consultations and approval of TSU. Messages on danger signs and the importance of their early identification and care seeking were disseminated using various channels. These included: putting up posters at all health facilities, wall paintings on

prominent walls of health facilities that are visible from the main roads, distribution of pamphlets during popular community fairs, addressing mothers and family members during antenatal care clinics, immunization day sessions and addressing community leaders during their meetings. These messages were also given by ASHAs during their routine HBNC visits.

## Quality control and assurance

**Activities by the health system.** ANMs supervised the daily activities of ASHAs through regular interaction, checking of registers and conducting home visits accompanied by ASHAs. The ANMs were supervised by the MOs at the PHCs. The work of the ASHAs, ANMs and MOs was reviewed by the BMO through monthly meetings and discussed in the district level meetings that are conducted on fixed days every month. These meetings were used as an opportunity to review the work done and conduct retrainings.

For the purpose of analysis, the 20-month implementation research duration was divided into three periods to represent the initiation of implementation, consolidation and full implementation respectively; first period: June to December 2017, second period: January to July 2018 and third period: August 2018 to January 2019. The performance of the health functionaries was monitored over these time periods.

**Activities by the research team.** The Program Learning team directly observed different processes, including interaction of health workers with families and the practice for management of sick young infants. The availability of adequate supplies (antibiotics, weighing scales, timers, thermometers, etc.), their use, storage facilities and utilization was checked monthly. Scheduled and random unscheduled visits were conducted at each facility to review the registers and documents maintained, for appropriateness and completeness. Specific training needs identified were addressed through targeted training sessions.

**Monitoring.** WHO monitors visited and interacted with research team and health functionaries. During these visits the progress was assessed, records of infants with PSBI were reviewed and practices of the health functionaries were observed.

**Data management.** Data was captured and entered in the web-based application Research Electronic Data Capture "REDCap" software. Data points were self-checked for completeness and transcription errors by the person filling the form. Logical and consistency checks were generated and resolved.

**Ethical considerations.** Ethical approvals were obtained from the CHRD-SAS Ethics Review Committee and Research Ethics Review Committee, WHO before initiation of the project activities. Written informed consent was taken from the families of young infants with PSBI, before collecting any information.

**Analysis methods.** For the quantitative analysis, frequencies and proportions were calculated for binary variables, proportion: of sick young infants identified, referred to a tertiary care hospital, that did not accept referral, treated at primary health facilities, completed treatment, compliance to treatment rates, death and other adverse outcomes using the STATA (R), 15.0 STATACORP, TX, USA software. The qualitative data was analysed using an inductive approach for examining the relationship between groups of data. The data were grouped using thematic schemes, narrative and content analysis.

## Results

### PSBI cases

During the study period (June 2017 to January 2019), 1997 live births were identified by the health workers (Table 1). ASHAs conducted ~ 75% of the scheduled post-natal HBNC home visits (Table 1).

**Table 1. Study population parameters.**

| Parameters | Number |
|---|---|
| Population of Sangrah Block | 77000 |
| Expected number of deliveries[1] | 2180 |
| Actual number of deliveries | 2010 |
| Total live births identified | 1997 |
| Number of Post-natal visits made by ASHA workers by Day[2] | n (%) |
| Day 1 (for home births) | 203 (57) |
| Day 3 | 1555 (78) |
| Day 7 | 1734 (87) |
| Day 14 | 1687(84) |
| Day 21 | 1606 (80) |
| Day 28 | 1498(75) |
| Day 42 | 1261(63) |

[1]Birth rate of 17/1000, 109 births per month for a total of 20 months

[2]For Day 1 visits N = 355 as this is applicable only for home births; for remaining visits N = 1977

Assuming a 10% incidence rate of PSBI among all live births [10–12, 22], we expected ~ 200 young infants with any sign of PSBI, in the population, including fast breathing in 7–59 day olds. We identified and treated 160 young infants with signs of PSBI, thus the coverage of treatment was 80% (Table 2).

Of the 160 PSBI cases, 18% (29/160) were classified as having critical illness (CI), 56% (92/160) as clinical severe infection (CSI), 3% (5/160) as severe pneumonia and 21% (34/160) as pneumonia.

A little less than half (75/160) of the PSBI cases were identified by families and brought to health facilities directly; over a fourth (45/160) were identified and/or referred to a health facility by the village level health workers (44 by ASHA and 1 by ANM). Forty of the 160 infants were identified in the hospital while the mother-infant pair was still hospitalized post-delivery. All the infants were healthy at birth, and all except two developed the symptoms of PSBI within 3 days after birth.

Referral to a hospital with pediatric care services was offered to 130 cases of PSBI by the MOs and ANMs. This included the 126 cases of CI, CSI and severe pneumonia (0–6 days) and 4 cases of pneumonia. Though referral of infants with pneumonia (only fast breathing in infants aged 7–59 days) is not recommended, the 4 cases were referred by the MO based on clinical judgement; however, 3 of them refused. Of the 130 cases that were referred, two-thirds (82/130) refused (Table 2).

Overall, 112 cases (82 who refused referral and 30 cases of pneumonia) were managed with the simplified antibiotic regimen. Of these, 58% (65/112) had CSI, 10% (11/112) had signs of CI, 3% (3/112) had severe pneumonia and 29% (33/112) had pneumonia (Table 2). Of the 33 infants with pneumonia who were managed with the simplified regimen, 7 were advised the recommended treatment of only oral amoxicillin, 19 were advised oral amoxicillin with injection gentamicin or another antibiotic and 7 were treated with antibiotics other than oral amoxicillin. Compliance to the full course of advised antibiotic treatment was 70% for oral amoxicillin i.e. 60 out of 86 advised amoxicillin, took all 14 doses and 63% for injection gentamicin i.e. 63 of the 101 advised gentamicin, got all 7 injections (Table 2).

Three-fourths (86/112) of the PSBI cases treated with the simplified antibiotic regimen were followed up on day 4 of treatment by the MOs and ANMs. Outcome ascertainment by

**Table 2. Identification, classification, treatment accepted, completion of treatment and treatment outcome for Infants 0–59 days with PSBI.**

| Parameters | Infant 0–59 days with PSBI | Infants 7–59 days of age with fast breathing only[4] (Pneumonia) | Infants 0–6 days with fast breathing only[3] (Severe pneumonia) | Infants 0–59 days with clinical severe infection[2] | Infants 0–59 days with critical illness[1] |
|---|---|---|---|---|---|
| **A. Identification** | **(n = 160)** | **(n = 34)** | **(n = 5)** | **(n = 92)** | **(n = 29)** |
| Born in health facility, healthy at birth and identified as having PSBI before discharge | 40 (25) | 0 | 3 (60) | 33 (36) | 4 (14) |
| Brought by families to health facility | 75 (47) | 20 (59) | 1 (20) | 35 (38) | 19 (65) |
| Identified by or referred to health facility by any community health workers | 45 (28) | 14 (41) | 1 (20) | 24 (26) | 6 (21) |
| By ASHAs | 44 (27) | 14 (41) | 1 (20) | 23 (25) | 6 (21) |
| By ANM | 1 (1) | 0 | 0 | 1 (1) | 0 |
| **B. Referral History** | **(n = 130)** | **(n = 4)[7]** | **(n = 5)** | **(n = 92)** | **(n = 29)** |
| Referral offered to a health care facility with pediatric services[5] | 130 | 4 | 5 | 92 | 29 |
| Refused referral | 82 (63) | 3 (75) | 3 (60) | 65 (71) | 11 (38) |
| Accepted referral | 48 (37) | 1 (25) | 2 (40) | 27 (29) | 18 (62) |
| **C. Simplified Treatment given at first level health facility (SC, PHC, CHC)[6]** | **(n = 112)** | **(n = 33)** | **(n = 3)** | **(n = 65)** | **(n = 11)** |
| Received 7 injections of gentamicin and 14 doses of suspension amoxicillin | 50 (45) | 16 (48) | 2 (67) | 34 (52) | 4 (36)[8] |
| **D. Compliance to treatment** | **(n = 112)** | **(n = 33)[9]** | **(n = 3)** | **(n = 65)** | **(n = 11)** |
| **Amoxicillin Suspension Received** | | | | | |
| 14 doses | 60 (53) | 17 (51) | 2 (67) | 37 (57) | 4 (36) |
| 10–13 doses | 13 (20) | 4 (12) | 0 | 5 (8) | 4 (36) |
| 1–9 doses | 13(20) | 5 (15) | 1 (33) | 5 (8) | 2 (18) |
| None (not advised) | 26 (40) | 7 (21) | 0 | 18 (28) | 1 (9) |
| **Additional antibiotics received** | 21 (19) | 7 (21) | 1 (33) | 12 (18) | 1 (9) |
| **Injection gentamicin Received** | | | | | |
| 7 injections | 63 (56) | 16 (48) | 3 (100) | 39 (60) | 5 (45) |
| 1–6 injections | 38 (58) | 10 (30) | 0 | 23 (35) | 5 (45) |
| No injection (not advised) | 11 (17) | 7 (21) | 0 | 3 (5) | 1 (9) |
| **E. Follow up** | N n (%) | N n (%) | N n (%) | N n (%) | N n (%) |
| Day 4 in all PSBI cases by health worker | 160 118 (74) | 34 20 (59) | 5 5 (100) | 92 70 (76) | 29 23 (79) |
| On Day 4 in cases who received simplified treatment given at first level health facility | 112 86 (77) | 33 19 (57) | 3 3 (100) | 65 55 (85) | 11 9 (82) |
| On Day 4 in cases who accepted referral to a hospital with pediatric care facility | 48 32 (67) | 1 1 (100) | 2 2 (100) | 27 15 (55) | 18 14 (78) |
| Day 8 follow up by research team for final outcome assessment | 160 159 (99) | 34 34 (100) | 5 5 (100) | 92 92 (100) | 29 28 (96) |
| **F. Outcome of the illness**<br>**i) In all PSBI cases identified** | **(n = 160)** | **(n = 34)** | **(n = 5)** | **(n = 92)** | **(n = 29)** |
| Recovered | 146 (91) | 29 (85) | 5 (100) | 86 (93) | 26 (90) |
| Treatment failure[10] | 14 (9) | 5 (15) | 0 | 6 (7) | 3 (10) |
| Severe adverse reaction | 2 (1) | 1 (3) | 0 | 1 (1) | 0 |
| Clinical deterioration | 2 (1) | 1 (3) | 0 | 1 (1) | 0 |
| Persistence | 4 (2) | 1 (3) | 0 | 3 (3) | 0 |
| Death | 6 (4) | 2 (6) | 0 | 1 (1) | 3 (10) |
| **ii) Outcomes in those who received simplified treatment at the first level health facility** | **(n = 112)** | **(n = 33)** | **(n = 3)** | **(n = 65)** | **(n = 11)** |
| Recovered | 103 (92) | 28 (85) | 3 (100) | 61 (94) | 11 (100) |
| Treatment failure[10] | 9 (8) | 5 (15) | 0 | 4 (6) | 0 |
| Severe adverse reaction | 2 (2) | 1 (3) | 0 | 1 (1) | 0 |
| Clinical deterioration | 1 (1) | 1 (3) | 0 | 0 | 0 |
| Persistence | 4 (3) | 1 (3) | 0 | 3 (5) | 0 |
| Death | 2 (2) | 2 (6) | 0 | 0 | 0 |
| **iii) Outcomes in those who accepted referral and were treated in a hospital with pediatric care facility** | **(n = 48)** | **(n = 1)** | **(n = 2)** | **(n = 27)** | **(n = 18)** |
| Recovered | 43 (89) | 1 (100) | 2 (100) | 25 (92) | 15 (83) |
| Treatment failure[10] | 5 (10) | 0 | 0 | 2 (7) | 3 (17) |
| Severe adverse reaction | 0 | 0 | 0 | 0 | 0 |
| Clinical deterioration | 1 (2) | 0 | 0 | 1 (4) | 0 |

*(Continued)*

**Table 2.** (Continued)

| Parameters | Infant 0–59 days with PSBI | Infants 7–59 days of age with fast breathing only[4] (Pneumonia) | Infants 0–6 days with fast breathing only[3] (Severe pneumonia) | Infants 0–59 days with clinical severe infection[2] | Infants 0–59 days with critical illness[1] |
|---|---|---|---|---|---|
| Persistence | 0 | 0 | 0 | 0 | 0 |
| Death | 4 (8) | 0 | 0 | 1 (4) | 3 (17) |

[1]Critical illness defined as having any of the following signs: Not able to feed, Convulsions, No movement

[2] Clinical severe infection defined as having any of the following signs: Stopped feeding well, movement only when stimulated, severe chest indrawing, Axillary Temp $\geq$ 37.5˚ C or < 35.5˚ C

[3]Severe Pneumonia defined as having: Fast Breathing, $\geq$ 60/min in age 0–6 days

[4]Pneumonia defined as having: Fast Breathing, $\geq$ 60/min in age 7–59 days

[5] Referral was offered to all cases of CSI, CI and SP to a hospital with pediatric care facility (DH/GH/Medical College). In addition, 4 cases of pneumonia were referred as per clinical judgement of the medical officer and 1 of them accepted referral

[6]Includes all cases who refused referral to a hospital with pediatric care facility and remaining 30 cases of pneumonia

[7]These children were offered referral as per clinical judgement by the medical officer though it is not recommended to refer these cases; 1 of them accepted referral and was treated in a higher facility; pre-referral dose was given and infant recovered.

[8]No infant with CI received Injection Ampicillin

[9]7 cases received only amoxicillin without any other injectable, 19 cases received inj. gentamicin along with amoxicillin, although they were not supposed to, 7 cases received injection gentamicin, but did not receive oral amoxicillin

[10]Treatment failure: An infant with PSBI with following outcomes: **Death**, **Clinical deterioration** defined as emergence of any sign of critical illness, any new sign of severe infection, need for care at higher level (hospitalization) during treatment, P**ersistence** defined as no improvement by day 4 or not fully recovered by day 8, **Severe adverse reaction** due to antibiotic administered as per simplified regimen

the research team was done in almost all infants with PSBI (159/160) on day 8 of treatment; one infant was away and could not be visited physically, but was confirmed to be alive by telephone. In the group treated with the simplified antibiotic regimen, by Day 8, 92% (103/112) of infants had recovered. Two infants, both classified as pneumonia, died after one day of treatment initiation; one each in the those treated with only oral amoxicillin and those receiving oral amoxicillin with injection gentamicin, giving a case fatality rate of 1.8%, (95% CI 0.2 to 6.3%) in this group (Table 2). In 5 cases (3 CSI, 2 pneumonia), treatment was initiated and completed by the ANMs at the SC, and all recovered.

Among those who accepted referral and were treated in a hospital with pediatric care services, 89% (43/48) recovered; there were 4 deaths, 3 in infants with CI and one with CSI. Case fatality rate in this group was 8.3%, (95% CI 2.3 to 19.9%) (Table 2). Two of these deaths occurred on the way to the hospital; both had been classified as CI and given pre-referral dose of injection gentamicin and oral amoxicillin by the MO.

## Information on live births from health workers

The RCH registers containing records of pregnancies and their outcomes maintained for each village by the ANMs were found to be incomplete. The primary reason was that ANM positions were vacant in almost half the SCs; the available ANMs were allotted additional work of the vacant SCs and were unable to do it well, with their other responsibilities. The HBNC home visitation registers maintained by ASHAs were also found to be incomplete. Around 80% of the ASHA workers said that this was so because they did not know how to fill some sections of the register or that it was time consuming, around 10% said the registers were not available, while 10% said the registers were not filled because they were unable to conduct

home visits during harsh weather conditions or due to poor access to homes. However, they did occasionally contact families over the phone to ask about the infant's health. Mothers of young infants also reported that ASHAs sometimes call them to enquire about their infants. Based on this feedback, the TSU facilitated a one-day training by the State trainer for the ASHAs, on filling the HBNC register.

The data entry of information collected on pregnancies, births and HBNC visits into the national Health Management Information System (HMIS) portal and the National Health Mission portal for Reproductive and Child Health (RCH) was delayed.

## Observations of PSBI cases managed with simplified treatment regimen

Direct observations by the research team of more than half (59 of 112) of the cases treated with the simplified regimen on an outpatient basis revealed that the MOs and ANMs checked the injection gentamicin vial for the expiry date, measured the advised amount correctly before administration, counselled the mother on care of injection site, explained the dose and administration of oral amoxicillin and when to return. In 58% (34/59) cases, infants were advised both antibiotics in the correct doses for their weight. Errors in calculating the dose were mostly for infants whose weight was at the upper limit of the weight band, given in the IMNCI chart booklet for calculating the dose [15]. Additionally, in two-thirds of the cases (44/59) the MOs and ANMs did not refer to the chart booklet to check weight specific dose. Errors in classification into CI, CSI, severe pneumonia and pneumonia of PSBI were observed in ~20% (37/160) of the cases; 25 infants were classified in a less severe category and 12 in a more severe category.16/59 observations were of cases classified as pneumonia and almost all (15/16) had also been advised injection gentamicin along with oral amoxicillin, which is not the recommendation.

During the three periods of monitoring (first period: Jun-Dec 2017, second period: Jan-Jul 2018, third period: Aug 2018-Jan 2019), the proportion of misclassification errors was 49% (17/37) in the first period, 30% in the second period (11/37) and 21% (8/37) in the third period. Of the total 21 errors in dosage of antibiotics (injection gentamicin), 57% (12/21), 38% (8/21), 5% (1/21) were made in the three periods, respectively; 19/21 (90%) resulted in under dosage of the infants (Table 3).

## In-depth interviews with different cadres of health workers during implementation

**Medical officers.** From the interviews conducted with 12 MOs, it was found that each examined around 10 young infants in a month; of which about three-fourths were brought directly by the families and one-fourth referred by an ASHA or ANM. Almost half of the families refused referral of sick young infants primarily due to high out-of-pocket expenditure (58%), non-availability of transportation (33%), family or social reasons (25%) or did not perceive illness to be severe (25%). The MOs found the community awareness activities very useful and suggested that these should also target the village leaders and local self-help groups.

**ANMs.** Interviews conducted with 14 ANMs revealed that they found the community awareness activities on recognition of danger signs and care seeking to be valuable. However, they could not be available at their assigned health facility because of other responsibilities. Only 20% could spontaneously recall all the signs and symptoms of PSBI correctly, while 70% recalled atleast 3 signs. Although, they were comfortable in using the IMNCI chart booklet for classification and dose calculation they were not confident in identifying PSBI cases, as they rarely got to see sick young infants.

Table 3. Periodic performance over time.

| Particulars | (N) | Period 1 (Jun-Dec,2017) | Period 2 (Jan-July,2018) | Period 3 (Aug,18-Jan,2019) |
| --- | --- | --- | --- | --- |
| | | n (%) | n (%) | n (%) |
| Live Births | 1997 | 598 (30) | 710 (35) | 689 (34) |
| PSBI Cases | 160 | 72 (45) | 51 (32) | 37 (23) |
| Sub Centers manned by ANMs | 27 | 16 (59) | 14 (52) | 11 (41) |
| Treatment initiated by ANM | 160 | 1 (1) | 1 (1) | 3 (2) |
| PSBI cases misclassified by health functionaries | 37 | 18 (49) | 11 (30) | 8 (21) |
| PSBI cases doses were inappropriate | 21 | 12 (57) | 8 (38) | 1 (5) |
| Under dosed for injection gentamicin | 19 | 10 (53) | 8 (42) | 1 (5) |
| Trainings | NA | 2[1] | 1 round of re-training in entire block, through GoI Videos[2] | - |
| District level meetings | 20 | 7 (35) | 7 (35) | 6 (30) |
| Block level meetings | 20 | 7 (35) | 6 (30) | 6 (30) |

[1] Respiratory rate measurement trainings of ASHAs & newly appointed MOs/ANMs training on PSBI Management.

[2] GoI Videos shown to all the MOs/ANMs/ASHAs.

**ASHAs.** During the interviews with 50 ASHA workers, 80% were unable to recall all the danger signs in young infants. Ninety percent expressed the need for re-training on recognition of danger signs, 80% on filling HBNC registers and 50% on counting of respiratory rate (RR). They felt that attending re-trainings on HBNC visits, using the job aids provided and participating in the conduct of the community awareness activities had improved their recognition as community health workers. They said their work was often interrupted because of non availability of functional thermometers, weighing scales, RR timers and HBNC registers. Also, they did not get timely incentives for completed work, which reduced their motivation.

**Families of infants with PSBI.** Of the 159 interviews with families of sick young infants during the day 8 visit, it was found that 10% used home remedies or treatment from traditional healers and unqualified practitioners before seeking care at a health facility, 81% sought care from a health facility within 24 hours of recognition of the illness. Five percent (7/159) sought care at a SC, 26% (42/159) at a PHC and 69% (110/159) at a CHC.

While ambulances and public transportation are available, only 30% of the families were able to avail these services as in certain parts of the block ambulances took 2–3 hours to arrive. In emergencies, people preferred to use private vehicles costing, at times, up to INR 2000–3000 (USD 30–40) one way to reach the health facilities.

## Quality of care at health facilities at time of initiation of the project

All PHCs and CHCs provided outpatient services for adults and children. Eighty percent (9/11) provided outpatient services for infants, 64% (7/11) had 24 hours emergency services and 10% (1/11) provided services for newborn care. Of the two CHCs in the block, CHC Sangrah provided regular inpatient, outpatient and delivery services for the entire block, however, there was a constraint of space, shortage of beds and the facility was not kept clean. There were periods of non availability of essential medicines and supplies ranging from a few days to a few months in a year. The CHC did not use any standard treatment protocol and there was no pediatrician or designated area for the care of neonates and young infants. The health centres had a structured referral form. Families who visited the health facilities were generally satisfied with the health workers behaviour as they were treated with kindness and compassion.

## Challenges and solutions

Some of the challenges identified during the implementation research and solutions instituted by the TSU are given in Table 4.

There were other barriers to implementing the simplified regimen that required solutions at a higher administrative and bureaucratic level. The health authorities acknowledged the shortage of manpower, but the process of hiring manpower was complicated and largely dependent on the political environment in the state. The ANMs that were appointed often got themselves transferred elsewhere due to the difficult terrain, unless they were locals. The availability of ANMs in the health facilities was neither consistent nor predictable as they had the additional charge of other SCs, community visits and meetings to attend. Ambulance availability was inadequate and only 30% of the families of infants were able to avail the ambulance

**Table 4. Challenges in implementation of the guidelines and solutions.**

| Health System level | Challenges | Solutions |
|---|---|---|
| **Identification of illness** | | |
| Community | Delayed recognition of illness and care seeking by community | Information, education and communication (IEC) material: Flip charts, posters, wall paintings depicting the signs and symptoms of PSBI, street play on PSBI, awareness sessions on PSBI during immunization and antenatal care days, village level local government meetings to increase recognition of illness and promote earlier care seeking closer to home |
| | | ASHAs informed families of the nearest functional health facility for prompt care seeking |
| | Small proportion of sick young infants identified | ASHAs were retrained on recognition of danger signs and essential newborn care |
| | | Job aids (handouts depicting signs and symptoms of PSBI) were provided to assist during home visits |
| Health Work Force | ASHAs not confident in counting respiratory rate | Respiratory rate timers provided; trained in counting respiratory rate; exercises to evaluate accuracy of respiratory rate count |
| **Confirmation and Treatment of PSBI** | | |
| Health Workforce | ANM not initiating or treating infants with PSBI | Encouragement and supportive supervision by MOs on a day to day basis; support from block and district health officials. Use other opportunities like monthly meetings for sharing experiences, success stories, resolving issues, allaying apprehensions and encouraging use of simplified treatment regimen. |
| | Errors in classification of PSBI and dose calculation | Regular retraining, targeted training on classification and dose calculation; GoI videos shown regularly |
| | | A chart on dose calculation put up in the outpatient area of all health facilities |
| Health Service Governance | Interrupted availability of MO and ANM at the health facility | AMOs trained in identification and management of PSBI |
| **Health System Infrastructure** | | |
| Health Products and Technologies | Non availability of ASHA HBNC registers | Facilitated procurement of the registers through the health system process |
| | Interrupted supplies of syringes, antibiotics, equipment | Supplies procured from government local purchase funds available at health centres Respiratory timers distributed to all health workers |
| | Maintenance of thermometers/lack of battery | Interacted with relevant MO to identify root cause in supply chain and raised issue during monthly meetings |
| Health Financing | Delayed payment of incentives for ASHAs | Escalated to district and state level and followed up actively with relevant authorities to hasten procurement and disbursement of incentives for ASHAs |
| Health Workforce | Challenges in completing RCH and HBNC registers by ANMs and ASHAs | Research team facilitated training in filling the forms and registers |

A few of the salient ones were:

a. To improve early recognition of illness and care seeking by the community, communication materials like flip charts and posters were provided and street plays on the importance of early identification of infants with PSBI were staged.

b. As there are fewer MOs, Ayurvedic Medical Officers (AMO) and practitioners of alternate medicines in the block were trained in identification and management of sick young infants.

c. To support health system infrastructure, procurement of supplies was facilitated and targeted trainings were provided.

service to reach the health facility when they identified the young infant to be sick or when they were referred to a hospital. A possible solution suggested to the health authorities was to provide more ambulances and increase the number of ambulance service hubs, so that their timely availability in all parts of the block is assured.

Injection ampicillin is part the essential drug list and should be available in the PHCs and CHCs, but was not available there. The option of buying the injection through the government funds available with the BMO, though suggested, could not be implemented as the MOs were reluctant to do so. Sick young infants with signs of critical illness, who refused referral, were instead treated with injection gentamicin and oral amoxicillin.

There was no standard process for requisition of supplies as well as equipment mainte-nance. For example, even procuring batteries for thermometers, which were provided to health workers in workshops or meetings conducted in a different PHC, was not possible simply because that instrument was not listed in the requisition book of that particular PHC.

## Discussion

Our implementation research demonstrated that it is possible to provide simplified treatment for sick young infants closer to home through the existing health system by providing adequate training, support and guidance to the health workers, as we reached our objective of 80% treat-ment coverage. Over 60% of the infants with PSBI were treated with the simplified antibiotic regimen through the primary health facilities in Sangrah. Thirty seven percent of families of sick young infants who accepted referral received treatment at a hospital that had pediatric care services, outside the block. The fact that all infants with PSBI received treatment is encouraging.

Coverage of our treatment was lower than that reported by the study in Zaria, Nigeria (96%) [23] but was higher than those reported from Malawi (64%) [24], Pune, India (57%) [25], Lucknow, India (53%) [26], Kushtia, Bangladesh (31%) [27]. Like our study, high treat-ment completion rates in infants with CSI treated on an outpatient basis were also reported by Malawi (95%) [24], Zaria, Nigeria (94%) [23] and MaMoni project, Bangladesh (80%) [28].

A large proportion of PSBI cases can be identified if families and community workers are taught to recognise illness and seek care early from an appropriate source; 80% of the esti-mated PSBI cases in the study area were identified, assuming 10% of infants would have sepsis [22]. Our 21% proportion of fast breathing pneumonia in 7–59 day-old infants was lower com-pared to 87% in Kushtia, Bangladesh [27], 49% in Pune, India [25], 27% in MaMoni project, Bangladesh [29], 28% in Malawi [24], 22% in Zaria, Nigeria [23] and higher compared to 13.3% in Lucknow, India [26]. Though it is possible that some cases of PSBI were missed, how-ever, an examination of the deaths reported in the state government portal, showed that of the 69 deaths among infants aged 0–59 days reported during the study period ~14 were due to probable sepsis. We were unable to identify any specific reason for the decreased proportion of PSBI cases identified in the second and third periods. A detailed quality check of all health facility records showed that no cases of PSBI captured there-in, were missed by the research team. Also, of the 14 deaths reported in the government portal that were due to probable sep-sis, half were in the first period and the other half over the second and third periods of our study.

The overall case fatality rate associated with PSBI in our setting was 3.7% (95% CI 1.4 to 7.9%) which was similar to Zaria, Nigeria (3.5%) [23] and Pune, India (3.4%) [25], higher than Malawi (0.3%) [24] and Lucknow, India (2.6%) [26] and lower than that reported in a meta-analysis in South Asia (8.7%; 95% CI 5.6 to 11.8%) [22]. Mortality in infants treated with the simplified regimen closer to home was lower than those who accepted referral to a hospital.

This finding is similar to that reported in other studies [11, 12, 23–26]. It must be noted, however, that a higher proportion of those who accepted referral had a more severe classification of illness.

The proportion with treatment failure among young infants with PSBI across all categories of illness, i.e. CI, CSI, severe pneumonia and pneumonia were similar among those treated with simplified regimen at the primary level health facilities compared to those treated at a hospital with pediatric care. These findings are similar to those from previous trials in Asia and Africa, where ~7–10% treatment failure was observed in cases of PSBI that were treated with the simplified antibiotic regimen [7, 10–12]. When compared with other PSBI implementation research studies, our treatment failure rate for CSI was 7% compared to 10.9% in Pune [25] and 5.8% in Lucknow, India [26], 3.5% in Malawi [24], 1.3% in Zaria, Nigeria [23]. However, our treatment failure rate for pneumonia patients (15%) was higher than that reported by Lucknow, India (4.8%) [26] and Malawi (2.2%) [24] and none in Pune, India [25] and Zaria, Nigeria [23].

More than three-fourths of the PSBI cases were followed up on day 4 by the health functionaries, as recommended in the guidelines, indicating a functioning health system even in this difficult terrain.

The process for identification of gaps by the program learning team, transparent feedback and prompt targeted retraining by the TSU helped in reducing subsequent errors made by the health system functionaries. Over time, errors in misclassification of PSBI reduced by more than 50%, while errors in giving inappropriate dosage of antibiotics decreased from 57% to 5%. Similar errors in misclassification and dosage have been reported from PSBI management from Pakistan [30] and MaMoni project [28], Bangladesh.

Awareness generation through community activities and empowering families of young infants to identify illness and seek care promptly in remote areas seems to be key to early identification of sick young infants with PSBI. In our setting, around half of the infants with PSBI were identified by families and brought to health facilities and the time to care seeking was reduced; over 80% sought care within 24 hours of illness recognition, which was an improvement from the observation during the baseline survey done in the same area, when only half of the families had sought care from a health facility within 24 hours of recognition of illness.

Families were the primary decision makers for care seeking and accepted care closer to home when trained personnel were available and referral was not feasible. High out of pocket expenditure, long distances, lack of prompt availability of ambulances or public transportation, family or social reasons and perception of illness being non-severe were some of the reasons identified for delayed care seeking or refusal to accept referral. The sub-district level hospitals about an hour away, did not have pediatricians posted which meant the closest health facility with a pediatrician was around 3 hours away from the block. These factors were also identified as common reasons for refusal to accept referral in other studies [24, 29, 31, 32]. During the baseline survey, no family had sought care at the SC level for sick young infants most likely because of non-availability of health workers at the SC. During this implementation research five cases of PSBI were treated by ANMs at the SC, three of which were after a year of initiation of the project, reflecting some positive change in the perception of families regarding availability of care at the Sub Center and the confidence of the ANMs in providing care. The fact that all recovered is reassuring. This was primarily possible because of the trainings and the encouragement of the MO who was in-charge of the ANMs who provided the simplified treatment. In these difficult to reach settings, the support and motivation of the MO played an important role in improving the performance of the health workers.

Adherence to the advised treatment was fairly high, which shows that the families had confidence in the treatment and that the infants improved with it. Other PSBI implementation research sites from Kushtia [27] and MaMoni [29] in Bangladesh, Ntcheu [24] in Malawi and

Zaria [23] in Nigeria also reported high adherence rate; whereas in Lucknow [26], India it was lower [23–24, 26–27, 29].

In one of the CHCs, which conducted all institutional deliveries, some cases of PSBI were identified in inborn infants on days 2 to 4 after birth while the mother and infant were still in the hospital. Mothers who delivered there were routinely discharged within 48 hours, but were sometimes kept for a longer duration for medical, social, economic and logistical reasons such as lack of transportation. It is unlikely that these infants had infection at birth or birth asphyxia, as they were assessed to be healthy at birth.

Even with the intensive training and support by the TSU, there were some gaps in the implementation. All 11 cases with critical illness that refused referral were treated with injection gentamicin and oral amoxicillin; none received injection ampicillin as recommended in the guidelines. Despite efforts by the research team, injection ampicillin could not be made available through the health system. A question that needs reflection by the policy makers is that in situations when it is challenging to procure the first line drugs what options or alternatives can be considered? With the issue of emerging antimicrobial resistance, careful consideration in selecting alternatives is important.

In this implementation research, infants aged 7–59 days with only fast breathing were to be treated with oral amoxicillin as decided during the national consensus meeting and training was also provided on the same [13]. However, only one fifth of such infants received only oral amoxicillin; the remaining received additional injectable antibiotics, because the national guidelines recommend using injectable gentamicin in these infants [3, 15]. The MOs therefore, seemed reluctant to give only oral antibiotics, despite trainings. This reluctance to follow guideline recommendations was also reported by sites from Bangladesh [27, 29]; Lucknow [26], India; and Pakistan [30], where, in many cases pre-referral antibiotic was also not given when indicated. Stronger advocacy by the state leadership in using the agreed regimen and behaviour change of the MOs is needed. Alignment of global, national and state level guidelines is also important.

Our experience affirmed that community level health workers like ASHAs can be trained to identify pneumonia and PSBI cases, as reported in other studies [8, 25, 26, 33]. Equipping ASHAs with skills through training in case identification and community sensitization led to better case detection. Improving their skills also gave them recognition in the community. The lower proportion of Day 1 visits for home births by ASHAs was primarily due to late information of birth or the women being at their maternal homes for delivery. Difficult terrain, inadequate public transportation, irregular incentives, interrupted supplies like lack of registers were other factors that prevented optimal performance. Regularizing the existing incentives to ASHAs for home visitation and connecting the incentives to identification of PSBI cases in the community could be helpful for motivation [16].

The delivery of the simplified regimen was implemented by the state leadership in one of the most difficult to reach areas in Himachal Pradesh in north India where timely hospitalization is not always feasible. The implementation was made possible by strengthening the existing public health system and empowering the health functionaries and local leadership. The TSU with partners from the state health authorities and research team played a critical role in providing guidance and support for the different activities, identification of training needs, imparting training, problem identification and suggesting possible solutions during the conduct of the implementation research. Negligible presence of private medical practitioners was another factor that facilitated the uptake of health initiatives through the public health system.

Our experience suggests that with initial support in training and help in developing systems it is possible to improve the functioning of the health system. The state of Himachal has a total population of around 7,400, 000 and 1,258,000 live births annually. Assuming a PSBI incidence

of ~10% and a case fatality rate of around 9%, approximately 11,500 deaths are expected among young infants annually [10, 11, 22]. With a case fatality rate of ~ 4%, as seen in this study, lives of approximately 6000 young infants can be saved in Himachal Pradesh by implementation of the simplified treatment regimen.

## Conclusion

Implementation of the simplified regimen for management of sick young infants with PSBI where referral is not feasible, through the existing public health care system, is possible and acceptable in Himachal Pradesh and can save lives of young infants. However, implementation and scale-up will require a) increasing awareness in mothers to identify illness and seek care early from appropriate sources, b) strengthening the routine post-natal home visitation program by the ASHAs, c) improving the skills and confidence of ANMs in providing care and providing them technical support through the MOs and other officials in the primary health care system, d) encouraging the MOs to follow the recommended guideline e) having uninterrupted availability of essential commodities and f) strengthening the referral transport systems. For implementation "one size fits all" may not hold true, understanding the local issues and barriers before implementation is crucial for the program to be successful.

## Supporting information

**S1 Form. List of pregnant women and home visit record.**
(PDF)

**S2 Form. Postnatal (HBPNC) home visit record.**
(PDF)

**S3 Form. Sick young infant assessment form.**
(PDF)

**S4 Form. Facility follow up form.**
(PDF)

**S5 Form. Sick newborn home visit.**
(PDF)

**S6 Form. Baseline health facility assessment form-CHC.**
(PDF)

**S7 Form. Baseline health facility assessment form- SC.**
(PDF)

**S8 Form. Baseline health facility assessment form-PHC.**
(PDF)

**S9 Form. Baseline health facility assessment form-district hospital/medical college.**
(PDF)

**S1 Questionnaire. Baseline survey mother questionnaire.**
(PDF)

**S2 Questionnaire. Baseline survey ASHA questionnaire.**
(PDF)

**S3 Questionnaire. Baseline survey in-depth mother questionnaire.**
(PDF)

**S1 IDI guide. In-depth interview: ANM.**
(PDF)

**S2 IDI guide. In-depth interview: ASHA.**
(PDF)

**S3 IDI guide. In-depth interview: MO.**
(PDF)

## Acknowledgments

The Centre for Health Research and Development Society for Applied Studies acknowledges the support provided by the Directorate of Health and Family Welfare, Himachal Pradesh (HP), India. We are grateful for the guidance and technical help provided by Dr. Anuj Gupta, Deputy Mission Director, National Health Mission, HP; Dr. Sanjay Sharma, Chief Medical Officer, District Sirmour, HP and Dr. Vinod Sangal, District Program Officer, Sirmour, HP. We are also thankful for the core support provided by the Department of Maternal, Newborn, Child and Adolescent Health, World Health Organization, Geneva (WHO Collaborating Centre IND-096). We acknowledge the contribution of Dr Ankit Raina (previously Sr Clinical Research Coordinator, CHRD-SAS) in supporting the trainings of all health functionaries in the block. We are also thankful to Kaushik Ghosh and Harsh Vardhan Jaiswal for their role in the development and commissioning of the REDCap software platform for data collection used in this implementation research.

## Author Contributions

**Conceptualization:** Shamim Ahmad Qazi, Samira Aboubaker, Yasir Bin Nisar, Rajiv Bahl, Maharaj Kishan Bhan, Nita Bhandari.

**Data curation:** Nidhi Goyal, Temsunaro Rongsen-Chandola, Bireshwar Sinha, Amit Kumar.

**Formal analysis:** Nidhi Goyal, Temsunaro Rongsen-Chandola, Bireshwar Sinha, Amit Kumar.

**Funding acquisition:** Shamim Ahmad Qazi, Samira Aboubaker, Yasir Bin Nisar, Rajiv Bahl.

**Investigation:** Nidhi Goyal, Temsunaro Rongsen-Chandola, Bireshwar Sinha, Amit Kumar.

**Methodology:** Nidhi Goyal, Temsunaro Rongsen-Chandola, Maharaj Kishan Bhan, Nita Bhandari.

**Project administration:** Nidhi Goyal, Temsunaro Rongsen-Chandola, Mangla Sood.

**Supervision:** Mangla Sood, Maharaj Kishan Bhan, Nita Bhandari.

**Writing – original draft:** Nidhi Goyal, Temsunaro Rongsen-Chandola, Bireshwar Sinha, Nita Bhandari.

**Writing – review & editing:** Nidhi Goyal, Temsunaro Rongsen-Chandola, Shamim Ahmad Qazi, Samira Aboubaker, Yasir Bin Nisar, Maharaj Kishan Bhan, Nita Bhandari.

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
