## [Decision Letter · Decision Letter 0]

15 Sep 2020

PONE-D-20-01536

Management of possible serious bacterial infection in young infants closer to home when referral is not feasible: Lessons from implementation research in Himachal Pradesh, India

PLOS ONE

Dear Dr. Rongsen-Chandola,

Thank you for submitting your manuscript to PLOS ONE. After careful consideration, we feel that it has merit but does not fully meet PLOS ONE’s publication criteria as it currently stands. Therefore, we invite you to submit a revised version of the manuscript that addresses the points raised during the review process.

We look forward to receiving your revised manuscript.

Kind regards,

Surbhi Leekha

Academic Editor

PLOS ONE

Journal Requirements:

2. Please include additional information regarding the surveys, questionnaires or interview guides used in the study and ensure that you have provided sufficient details that others could replicate the analyses. If you developed and/or translated surveys, questionnaires or interview guides as part of this study and they are not under a copyright license more restrictive than Creative Commons Attribution (CC-BY), please include a copy, in both the original language and English, as Supporting Information.

5. Please ensure that you refer to Figure xxxxx in your text as, if accepted, production will need this reference to link the reader to the figure.

Reviewers' comments:

Reviewer's Responses to Questions

**Comments to the Author**

1. Is the manuscript technically sound, and do the data support the conclusions?

Reviewer #1: Yes

Reviewer #2: Partly

2. Has the statistical analysis been performed appropriately and rigorously? 

Reviewer #1: Yes

Reviewer #2: I Don't Know

3. Have the authors made all data underlying the findings in their manuscript fully available?

Reviewer #1: Yes

Reviewer #2: Yes

4. Is the manuscript presented in an intelligible fashion and written in standard English?

Reviewer #1: Yes

Reviewer #2: Yes

5. Review Comments to the Author

Reviewer #1: Goyal et al., set out to document their findings on the implementation of the 2015 WHO guidelines on the management of possible serious bacterial infections where referral is not feasible in a select community in Himachal Pradesh, India. The work is highly significant with potential for huge impact in management of sick young infants with sepsis in the developing world. Their strategy is well thought out. However, there are some issues that the team needs to think about:

• Was the work based on particular implementation science theoretical framework and if yes can they briefly outline these to enable readers easily understand their contextual approaches from inputs to outcomes.

• They have done well to look at health system level barriers such as availability of commodity/supplies and infrastructure, however, there are other health system level issues (so called blocks) such as Service Delivery Systems (Health service organization); Leadership and Governance (health service management); Health Workforce (Human Resources for the provision of Health Services); Health Financing (Resources for service provision); Health Products and Technologies (essential medicines, medical supplies, vaccines, health technologies, and public health commodities required in provision of services); Health Information (Systems for generation, analysis, dissemination, and utilization of health-related information); Health Infrastructure (physical infrastructure, equipment, transport, and Information Communication Technology needed) which would impact on the implementation of the PSBI guidelines. However, these will have best been evaluated through organizational capacity index (assessment). What, is their estimation of how these factors interact in their setting to impact PSBI guidelines implementation activities?

• Table 2, section F under clinical deterioration, there is no n value provided whilst there is a percentage value of 3%

• Only 20% of ANMs could spontaneously recall PSBI signs and symptoms. Recommendations on training? How can their role in PSBI management be better reinforced? There is need to identify how ANMs and AMOs can further complement other health functionaries in PSBI management.

• Is there a checklist on identification of danger signs in SYI that ASHAs utilize while conducting home visits?

• The percentage of those who refused referral is quite high at 63%, and warrants a brief discussion of the main causes for this.

• Only 30% of families could access referral services yet this was not highlighted as a critical challenge. Solutions should also be proposed on a strengthened referral pathway as a vital pillar in PSBI implementation.

• While the study has a well-documented scenario of quality improvement in service delivery around PSBI management, there is need to document the existing health system quality improvement strategy that can be leveraged in future to ensure skills retention so as to avoid challenges such as misclassification of PSBI especially with new staff deployments.

• The study raises key issues in availability of ampicillin in PSBI guidelines implementation. This is a critical lesson in implementation of PSBI and the availability of essential medicines such as benzylpenicillin as alternatives. However, a different question of antimicrobial resistance in the case of a restricted drug such as ceftriaxone should be considered.

Reviewer #2: The manuscript could benefit from some alignment between the objectives of the project as stated on pages 3 and 4 and the research methods used to achieve these objectives. In particular, the "encourage" objective was not well supported with the interventions used (mainly training and Feedback).

There was no mention of how data systems at all levels captured the simplified treatment related data when the treatment was offered and whether data at the ANMs and ASHAS were integrated with the larger health systems for improvement purposes.

In examining quality of delivery of the simplified treatment, authors did not indicate whether there was an inter-rater reliability established.

Also, there was no description of the simplified treatment intervention and whether fidelity measures for its use were established as these are essential for quality assessment- which the authors indicated that they have completed.

Qualitative research instruments were not included, but they were the bases for the recommendations made on how to overcome some of the barriers for offering the simplified treatment.

6. PLOS authors have the option to publish the peer review history of their article (what does this mean?). If published, this will include your full peer review and any attached files.

Reviewer #1: **Yes: **Dr Jesse Gitaka

Reviewer #2: No

---

## [Author Response · Author response to Decision Letter 0]

30 Oct 2020

Authors Response: Thank you. The manuscript has been prepared as per PLOS ONE style requirements. 

2. Please include additional information regarding the surveys, questionnaires or interview guides used in the study and ensure that you have provided sufficient details that others could replicate the analyses. If you developed and/or translated surveys, questionnaires or interview guides as part of this study and they are not under a copyright license more restrictive than Creative Commons Attribution (CC-BY), please include a copy, in both the original language and English, as Supporting Information.

Authors Response: Thank you. We have now included the data collection forms and instruments used for surveys, health facility assessments and in-depth interviews as supporting information

Authors Response: Thank you. Data is available with the primary author and will be made available when the manuscript is accepted for publication. This has been mentioned in the cover letter. 

Authors Response: We have validated the existing ORCID id in the Editorial Manager. 

5. Please ensure that you refer to Figure xxxxx in your text as, if accepted, production will need this reference to link the reader to the figure.

Authors Response: Thank you. Reference of the relevant figures (Figure number) has been added and updated in the applicable section of the manuscript. 

Reviewers' comments:

Reviewer's Responses to Questions

Comments to the Author

1. Is the manuscript technically sound, and do the data support the conclusions?

Reviewer #1: Yes

Reviewer #2: Partly

2. Has the statistical analysis been performed appropriately and rigorously? 

Reviewer #1: Yes

Reviewer #2: I Don't Know

3. Have the authors made all data underlying the findings in their manuscript fully available?

Reviewer #1: Yes

Reviewer #2: Yes

4. Is the manuscript presented in an intelligible fashion and written in standard English?

Reviewer #1: Yes

Reviewer #2: Yes

5. Review Comments to the Author

Reviewer #1: Goyal et al., set out to document their findings on the implementation of the 2015 WHO guidelines on the management of possible serious bacterial infections where referral is not feasible in a select community in Himachal Pradesh, India. The work is highly significant with potential for huge impact in management of sick young infants with sepsis in the developing world. Their strategy is well thought out. However, there are some issues that the team needs to think about:

• Was the work based on particular implementation science theoretical framework and if yes can they briefly outline these to enable readers easily understand their contextual approaches from inputs to outcomes.

Authors Response: Thank you. We adapted the RE-AIM framework for developing the conceptual framework for the study. The framework has been included in the manuscript as figure 1 and is briefly outlined in lines 91-94. 

• They have done well to look at health system level barriers such as availability of commodity/supplies and infrastructure, however, there are other health system level issues (so called blocks) such as Service Delivery Systems (Health service organization); Leadership and Governance (health service management); Health Workforce (Human Resources for the provision of Health Services); Health Financing (Resources for service provision); Health Products and Technologies (essential medicines, medical supplies, vaccines, health technologies, and public health commodities required in provision of services); Health Information (Systems for generation, analysis, dissemination, and utilization of health-related information); Health Infrastructure (physical infrastructure, equipment, transport, and Information Communication Technology needed) which would impact on the implementation of the PSBI guidelines. However, these will have best been evaluated through organizational capacity index (assessment). What, is their estimation of how these factors interact in their setting to impact PSBI guidelines implementation activities?

Authors Response: Thank you. We agree with the esteemed reviewer that all health system level issues are important and would impact the implementation of the PSBI guidelines. We have looked at some of these aspects and the applicable health system level for each of the identified challenges has now been added to the manuscript in Table 4: Challenges in implementation of the guidelines and solutions. 

In our estimation implementation at the individual and intervention level was affected by sub-optimal functioning of factors at the higher system level. For example, as mentioned in the manuscript, i) although the ANMs were trained to initiate the simplified treatment regimen, there interrupted availability at the health centers due to the shortage of manpower meant that they had very few opportunities to initiate the simplified regimen. ii) The health functionaries were aware of the antibiotic regimen to be used, but non-availability of certain antibiotics, because of procurement issues affected the implementation. iii) There were delays in replacement of batteries for thermometers due to the supply chain system. iv) The routine entry of data collected and maintained by the ASHAs and ANMs into the National Health Management Information System (HMIS) portal and the National Heath Mission Portal for Reproductive and Child Health (RCH) was delayed due to lack of adequate health workforce and infrastructure. 

• Table 2, section F under clinical deterioration, there is no n value provided whilst there is a percentage value of 3%

Authors Response: Apologies for the omission. The value should be 1; it has been updated in the table.

• Only 20% of ANMs could spontaneously recall PSBI signs and symptoms. Recommendations on training? How can their role in PSBI management be better reinforced? There is need to identify how ANMs and AMOs can further complement other health functionaries in PSBI management.

Authors Response: Thank you for your observation. At our site, while only 20% of ANMs could spontaneously recall all signs and symptoms of PSBI, 70% were able to spontaneously recall at least 3 signs, as assessed around 6 months after initiation of the project (added in the manuscript in line 425). Though trained, they had very few opportunities to examine sick young infants and use their knowledge practically because of their unpredictable availability at the health centres due to inadequate human resources (mentioned in the manuscript in lines 471-473).

Some of the solutions implemented to address this include retraining, targeted training, providing job-aids like flip charts and posters with pictorial representation of danger signs and putting up posters in the health centers with the treatment algorithms and dosage. As anticipated, encouragement and supportive supervision of the ANMs by the MOs was very important. In addition, appointment of ANMs against the vacant positions would have improved availability of ANMs at the health centres increasing their opportunity to examine infants and initiate the simplified regimen. However, as mentioned in the manuscript in lines 468-470, this was not possible as the process of hiring manpower is protracted and complicated. 

This study showed that with strengthening of their skills and re-tasking their roles, ANMs and AMOs could identify and treat sick young infants with PSBI and support the health system in partly overcoming the shortage of trained manpower.

• Is there a checklist on identification of danger signs in SYI that ASHAs utilize while conducting home visits?

Authors Response: Yes. The GOI home based newborn care operational guidelines (Reference number 17) and the ASHA home visitation registers have the listing of danger signs for identification of SYI for use by ASHAs. As mentioned in the manuscript lines 262-264, all ASHAs and ANMs were also provided flip charts with pictorial representation of danger signs, elements of essential newborn care and messages for the family for care seeking, as part of the implementation strategy.

• The percentage of those who refused referral is quite high at 63%, and warrants a brief discussion of the main causes for this.

Authors Response: This implementation research was purposively conducted in a unique setting with difficult terrain from where referral is difficult or not always feasible. In our setting “refused referral” also included families who could not accept referral due to reasons like bad weather or blocked roads due to landslides, besides those who refused due to personal or social reasons. The overall percentage of families who refused referral was high. However, among those who had critical illness 62% accepted referral; the higher proportions of refusals were in those with relatively milder forms of illness. 

The main causes for the high proportion of those who refused referral include high out of pocket expenditure, long distances, lack of prompt availability of ambulances or public transportation, family or social reasons and perception of illness being non-severe were some of the reasons identified for delayed care seeking or refusal to accept referral to a hospital. In addition, the sub-district level hospitals, that are about an hour away, did not have pediatricians posted in them and the closest health facility with a pediatrician was around 3 hours away from the block. These factors have been identified by other studies as common reasons for refusal to accept referral , , , . This has been revised in the manuscript in lines 548-554. 

• Only 30% of families could access referral services yet this was not highlighted as a critical challenge. Solutions should also be proposed on a strengthened referral pathway as a vital pillar in PSBI implementation.

Authors Response: Thank you. While ambulance service and public transportation is available only 30% of the families were able to avail these services as in certain parts of the block ambulances took 2-3 hours to arrive. In emergencies people preferred to use private vehicles costing, at times, up to INR 2000-3000 (USD 30-40) one way to reach the health facilities.

A possible solution shared with the health authorities was to provide more ambulances and increase the number of ambulance service hubs, so that their timely availability in all parts of the block is assured.

 We have added this in the manuscript in lines 441-444 and lines 475-477. 

• While the study has a well-documented scenario of quality improvement in service delivery around PSBI management, there is need to document the existing health system quality improvement strategy that can be leveraged in future to ensure skills retention so as to avoid challenges such as misclassification of PSBI especially with new staff deployments.

Authors Response: Thank you. The skill retention and quality improvement strategy that was implemented within the existing health system consisted of the following: 

i) Health functionaries from within the system were trained as trainers and they conducted all the trainings and re-trainings for the health workers, physicians and other cadres on management of sick young infants. The intention was to equip the health system to be able to continue implementation of the guidelines and its scale up even after the study ended. 

ii) Yearly retraining sessions, by the state trainers, through in-person trainings and showing videos; and use of aids like posters with treatment algorithms and dosages of antibiotics, in health centres was done to support skills retention. 

iii) The health functionaries from Sangrah block shared their experiences and learnings from the implementation of the guidelines at their health centres with the health officials from the neighbouring blocks and districts by way of organizing trainings and visits to Sangrah block to observe the implementation.

iv) The work of the grass root level workers like ASHA and ANMs in the block is reviewed through monthly meetings by the PHC and CHC MOs. This opportunity was leveraged to conduct regular retraining sessions and review quality. This platform also provided an opportunity for the officials and trainers to share experiences and success stories, discuss problems faced and how they were overcome, allay apprehensions of the health staff and encourage them to use the simplified treatment regimen. 

v) The monthly antenatal care day (ANC), immunisation day and the ASHA Home Based Neonatal Care visits (HBNC) provided opportunities to interact with the community elders and young and expectant mothers. These were used to increase awareness on identification of sick young infants and seeking care and reaching out to the ASHA/ANMs as early as possible.

While we recognize that stock out of the essential antibiotics, mentioned in the manuscript in lines 479-482 is an important quality of care issue, it could not be addressed satisfactorily during the implementation as the problem was multifarious resulting from a combination of supply chain and individual will factors and would need addressing within the system. 

These points have been updated in the manuscript in lines 276-277, 282-284 and Table 4 Challenges in implementation of the guidelines and solutions.

• The study raises key issues in availability of ampicillin in PSBI guidelines implementation. This is a critical lesson in implementation of PSBI and the availability of essential medicines such as benzylpenicillin as alternatives. However, a different question of antimicrobial resistance in the case of a restricted drug such as ceftriaxone should be considered.

Authors Response: Thank you. We agree that it is critical to ensure uninterrupted availability of essential medicines. In particular situations where the essential medicines are not readily available, the choice and use of other important antimicrobial drugs should be made after careful consideration, in view of emerging antimicrobial resistance. We have added this in the manuscript in lines 577-578.

Reviewer #2: The manuscript could benefit from some alignment between the objectives of the project as stated on pages 3 and 4 and the research methods used to achieve these objectives. In particular, the "encourage" objective was not well supported with the interventions used (mainly training and Feedback).

Authors Response: Thank you. We apologise for not presenting the methods in a way that clearly describes how the objectives were achieved. For this study, the implementation was through the health system and research team supported and encouraged the health functionaries to use the simplified treatment regimen as follows: 

a) The Technical Support Unit (TSU) comprising of the state health officials, including the Deputy Mission Director - National Health Mission, State Programme Officer - Child Health, District Programme Officer - Sirmaur, Chief Medical Officer - Sirmaur and the researchers of CHRD-SAS was established to encourage ownership of the implementation of intervention by the state. 

b) Capacity building was done through supporting training and identifying targeted training needs; providing job aids like flip charts, IMNCI chart booklet and posters; support by providing guidance in documentation; explaining to the health staff the advantages of early identification and understanding their challenges in managing PSBI cases. 

c) Effective use of opportunities: Monthly meetings conducted at the primary health center, block and district level on fixed days every month were used as a platform to conduct regular retraining sessions and review quality. These also provided the officials and trainers an opportunity to share experiences and success stories, discuss issues and how they were overcome and allay apprehensions of the health staff and encourage them to use the simplified treatment regimen.

 These have been more clearly articulated in the manuscript in lines 174-176, 266-269, 283- 284 and Table 4 Challenges in implementation of the guidelines and solutions. 

There was no mention of how data systems at all levels captured the simplified treatment related data when the treatment was offered and whether data at the ANMs and ASHAS were integrated with the larger health systems for improvement purposes.

Authors Response: Thank you. We have incorporated the information on how the data was captured at the different levels within the health system and from where the research team captured this information for the study, in the manuscript section on data collection instruments and procedures lines 216-237. 

The data collected and maintained by the ASHAs and ANMs is routinely entered into the national Health Management Information System (HMIS) portal and the National Health Mission portal for Reproductive and Child Health (RCH). The data entry into these portals is sometimes delayed and incomplete (added in the manuscript in lines 388-390). In the monthly review meetings at the PHC / Block level, health related data maintained by the ASHA/ANMs is collated and reviewed. This provides an opportunity to review information on management of PSBI cases in the area periodically for improvement purposes. During the project, the errors or issues faced by ASHAs and ANMs related to data entry and maintenance were identified and shared with their supervisors at the block and district level monthly meetings for the purpose of improvement.

In examining quality of delivery of the simplified treatment, authors did not indicate whether there was an inter-rater reliability established.

Authors Response: We conducted pre and post training assessments to assess knowledge, skill retention and further training needs for all health functionaries (mentioned in lines 254-255). The intention was to develop an implementation strategy that would replicate the actual program conditions in the public health system where the usual practice is to provide training to the health functionaries and for them to implement the program. 

Additionally, as part of capacity building and skill assessment, evaluation of the accuracy of respiratory rate count in young infants by ASHAs, post training, was compared to a gold standard (trainer). This was done because the ASHA workers are the first contact between the SYI and the health system and identification and referral of PSBI or simple fast breathing can be done earliest by them. 

The exercise showed strong corelation between RR measurements by ASHA and the gold standard with Pearson correlation coefficient >0.7 (P=0.001). The mean difference between the readings by ASHAs as compared to Gold standard was -0.74 (-1.9; 0.41). Bland-Altman plot was constructed to study the limits of agreement. Analysis revealed acceptable agreement between the RR count between ASHAs and the Gold standard in infants.

This has been added in the manuscript in lines 256-260.

Also, there was no description of the simplified treatment intervention and whether fidelity measures for its use were established as these are essential for quality assessment- which the authors indicated that they have completed.

Authors Response: Thank you for your comment. The simplified treatment intervention included Oral amoxicillin twice daily (50mg/kg/dose) plus intramuscular injection gentamicin once a day (5mg/kg) for 7 days for infants with clinical severe infection or severe pneumonia; and only oral amoxicillin twice daily (50mg/kg/dose) for 7 days for those with pneumonia. This is given in Figure 3 of the manuscript. 

Fidelity of the implementation of the adapted guidelines for management of PSBI was measured through direct observations of the process and review of the records maintained at the health centers by the health functionaries. It was noted that for fast breathing alone, the simplified treatment regimen of only oral amoxicillin was not being followed (mentioned in the manuscript lines 350-352 and 355-358). The reasons have been discussed in the manuscript lines 579-588.

Qualitative research instruments were not included, but they were the bases for the recommendations made on how to overcome some of the barriers for offering the simplified treatment.

Thank you for pointing this out. We have now included the data collection forms and instruments used for surveys, health facility assessments and in-depth interviews as supporting information.

6. PLOS authors have the option to publish the peer review history of their article (what does this mean?). If published, this will include your full peer review and any attached files.

Do you want your identity to be public for this peer review? For information about this choice, including consent withdrawal, please see our Privacy Policy.

Reviewer #1: Yes: Dr Jesse Gitaka

Reviewer #2: No

---

## [Editor Report · Decision Letter 1]

30 Nov 2020

Management of possible serious bacterial infection in young infants closer to home when referral is not feasible: Lessons from implementation research in Himachal Pradesh, India

PONE-D-20-01536R1

Dear Dr. Rongsen-Chandola,

We’re pleased to inform you that your manuscript has been judged scientifically suitable for publication and will be formally accepted for publication once it meets all outstanding technical requirements.

Kind regards,

Surbhi Leekha

Academic Editor

PLOS ONE
---

## [Editor Report · Acceptance letter]

7 Dec 2020

PONE-D-20-01536R1 

Management of possible serious bacterial infection in young infants closer to home when referral is not feasible: Lessons from implementation research in Himachal Pradesh, India 

Dear Dr. Rongsen-Chandola:

I'm pleased to inform you that your manuscript has been deemed suitable for publication in PLOS ONE. Congratulations! Your manuscript is now with our production department. 

Kind regards, 

on behalf of

Dr. Surbhi Leekha 

Academic Editor

PLOS ONE